# Identifying Strategies to Mitigate Cybersickness in Virtual Reality Induced by Flying with an Interactive Travel Interface

Daniel Page [ID], Robert W. Lindeman [ID] and Stephan Lukosch *[ID]

HIT Lab NZ, University of Canterbury, Private Bag 4800, Christchurch 8140, New Zealand;
daniel.page@pg.canterbury.ac.nz (D.P.); rob.lindeman@canterbury.ac.nz (R.W.L.)
* Correspondence: stephan.lukosch@canterbury.ac.nz

**Abstract:** As Virtual Reality (VR) technology has improved in hardware, accessibility of development and availability of applications, its interest has increased. However, the problem of Cybersickness (CS) still remains, causing uncomfortable symptoms in users. Therefore, this research seeks to identify and understand new CS mitigation strategies that can contribute to developer guidelines. Three hypotheses for strategies were devised and tested in an experiment. This involved a physical travel interface for flying through a Virtual Environment (VE) as a Control (CT) condition. On top of this, three manipulation conditions referred to as Gaze-tracking Vignette (GV), First-person Perspective with members representation (FP) and Fans and Vibration (FV) were applied. The experiment was between subjects, with 37 participants randomly allocated across conditions. According to the Simulator Sickness Questionnaire (SSQ) scores, significant evidence was found that GV and FP made CS worse. Evidence was also found that FV did not have an effect on CS. However, from the physiological data recorded, an overall lowering of heart rate for FV indicated that it might have some effect on the experience, but cannot be strongly linked with CS. Additionally, comments from some participants identified that they experienced symptoms consistent with CS. Amongst these, dizziness was the most common, with a few having issues with the usability of the travel interface. Despite some CS symptoms, most participants reported little negative impact of CS on the overall experience and feelings of immersion.

**Keywords:** virtual reality; cybersickness; travel interface; field of view; first-person perspective; multi-sensory

## 1. Introduction

Despite improvements in Virtual Reality (VR) technology, there still remains a problem with its usability. During usage, people can experience a variety of discomforting symptoms, often referred to as Cybersickness (CS), VR sickness or Visually Induced Motion Sickness (VIMS). These include nausea, disorientation, oculomotor disturbances, drowsiness, eye strain, headache, pallor, sweating, dry mouth, fullness of stomach, vertigo, ataxia and emesis [1,2]. Longer exposure to VR has been found to produce more of these symptoms [3–5]. While it seems the majority of people can tolerate prolonged exposure, there is a small number of users that experience an emetic (retching/vomiting) response [6–9]. Effects have also been found to remain after exposure (varying between 10 min and four hours) [10], compromising postural stability, hand-eye coordination, visual functioning and general well-being [11]. However, it has been found that users can develop a tolerance over repeated sessions [3,12]. CS has similarities to Motion Sickness (MS) but is different since it is caused by visually induced motion [2]. Simulator Sickness (SS), which came about with the invention of HMDs [10], is another type of sickness arising from the use of simulators and is closely related to CS. These terms are often used interchangeably [11].

It is useful for developers to have guidelines for designing usable and pleasant VR experiences that avoid discomfort. While some guidelines do exist, CS is not fully understood [13], and there are still aspects that need to be studied (2017 Oculus Best Practice

Guide: https://static.oculus.com/documentation/pdfs/intro-vr/latest/bp.pdf (accessed on 15 March 2023)) [14]. For example, it varies significantly between users and is difficult to quantify precisely [8,15]. Mitigating CS is particularly important for an increasing number of applications with different types of movement, interaction, interfaces for travelling and sensory inputs that seek to heighten immersion. Therefore, based on this need, our work intends to identify novel strategies for CS mitigation.

Techniques manipulating Field of View (FOV) have been tried with some success. However, these can constrain the visual experience and reduce immersion [12,16–18]. A strategy we propose, which builds upon other FOV-related strategies while aiming to provide full immersion, is dynamically manipulating FOV in VR with a vignette based on virtual speed and eye-gaze.

Having a first-person perspective in VR is common, and it has been suggested that it affects postural stability, which has been linked with CS [1,19,20]. However, there is little variation in the extent of member representation and virtual movement in the literature [19]. We propose a strategy that provides a close approximation of a first-person perspective with member representation.

Physical and virtual conflict is the most common CS theory. Studies have attempted to reduce this conflict using different types of sensory stimulation [21]. However, these are not always well matched to the experience with multiple senses stimulated. We propose simulating vibration and airflow that closely match a VR experience.

These proposed strategies were formed into four conditions called Control (CT), Gaze-tracking Vignette (GV), First-person Perspective with members representation (FP) and Fans and Vibration (FV). To test each, a cybersickness-inducing experience was created involving a novel physical "flying" broomstick travel interface to control its virtual counterpart through a Virtual Environment (VE). During each experimental session, the level of CS was assessed using objective and subjective measurements. The data was then analysed using statistical methods.

## 2. Background

There are different ideas about what the underlying causes of CS or closely related sicknesses are, and what indicates whether a person will be susceptible to an experience. Two of the predominant theories are introduced in this section.

### 2.1. Theories on what Causes Cybersickness

The most widely cited theory [1] is the *sensory conflict hypothesis*, which postulates that CS is due to sensory conflicts in expected signals from visual, auditory, tactile, kinaesthetic and vestibular activity [1,22]. One explanation for this is that susceptibility to CS is associated with an individual's ability to quickly reweight multisensory cues that conflict [1,22].

The *postural instability hypothesis* postulates that MS is caused by alterations in postural control. Postural instability arises when new dynamics in an experience are failed to be perceived and controlled with appropriate actions [1,20].Chardonnet et al. [23] proposed that when exposed to 3D stimuli in VR, a physiological mechanism called the "righting reflex" does not work as well and can increase postural sway.

### 2.2. Factors That Influence Cybersickness

Several factors that have been reported to influence CS and closely related sicknesses [14] are examined in this section.

#### 2.2.1. Posture and Embodiment

Studies have found evidence suggesting that a sitting posture can be beneficial for reducing CS [20,24,25]. However, just sitting down does not appear to prevent users from becoming sick [26]. Riccio and Stoffregen [20] note that standing might be related to

increased postural instability, which leads to increased levels of MS. However, research suggests that postural sway is not necessarily a prerequisite for CS [27].

Debarba et al. [19] also conducted an experiment that involved manipulating the amount of visible embodiment (body ownership) that a user sees in VR. The Simulator Sickness Questionnaire (SSQ) was used to see if there was an effect on SS, but they found no significant difference between conditions. However, it should be noted that this testing did little to induce CS with users remaining seated. Additionally, it has been argued that the degree of embodied interaction provided by a locomotion interface can impact CS [25].

### 2.2.2. Field of View

Humans have a total horizontal FOV of about 180°, with a 120° overlap region and 30–35° unique to each eye [28]. There is much variation in the FOV of different VR HMDs which range from about 90° to 210° horizontally (Virtual Reality Devices: https://xinreality.com/wiki/Virtual_Reality_Devices (accessed on 15 March 2023)). It has been found that as the horizontal FOV increases, so do the symptoms of CS and the level of presence. However, above 140° the difference in symptoms has not been found to be significant [28].

### 2.2.3. Flicker

Jarring changes in brightness, referred to as *display flicker*, can occur in VR experiences. This impacts the oculomotor component of SS, leading to eye strain, fatigue or headaches. However, people have different sensitivities to it. It is perceived most strongly in the periphery of the FOV and is worse with bright or high-contrast content, rapidly alternating content, fine patterns and a low refresh rate. OLED displays can also contribute to some degree of flicker (2017 Oculus Best Practice Guide) [29–31].

### 2.2.4. Virtual Movement and Altitude

The rate of the onset of SS has been identified as being proportional to the speed of navigation [32]. While at constant speeds, no vestibular response is expected, during acceleration there is. According to the sensory conflict theory, this leads to discomfort. Specifically, the 2017 Oculus Best Practice Guide notes that the severity of discomfort is a product of the frequency, size and duration of acceleration. Another related factor is altitude; when a user is virtually close to the ground, it tends to fill their FOV creating a more intense optical flow during movement (2017 Oculus Best Practice Guide). This could have some interaction with speed or acceleration on the severity of CS symptoms.

### 2.3. Strategies Applied to Mitigate Cybersickness

Strategies from the literature that have been applied to mitigate the symptoms of CS and the closely related SS are outlined below.

### 2.3.1. FOV Manipulation

Fernandes and Feiner [12] conducted an experiment in which FOV was dynamically adjusted using a vignetting technique. This was intended to be subtle to avoid decreasing the sense of presence and to minimise the user's awareness of the intervention. When there was a mismatch between the physical and virtual motion, the FOV was reduced, but when both were consistent, the FOV was normalised. In a study, they found evidence to suggest that this technique reduced the degree of VR sickness and helped participants adapt to VR. Norouzi et al. [16] found that vignetting with amplified head rotations as an input caused participants to experience significantly more VR sickness. However, this is in contrast to similar work involving vignetting that used controller-based input [16].

### 2.3.2. Resolving Sensory Conflict with Vibration

Jung et al. [21] conducted a study that aimed to reduce CS by reducing sensory conflict. This involved a vibrating floor that delivered vestibular stimuli, matching the expected

vibration characteristics of an off-road driving simulation designed to be CS-inducing. The results of this study found evidence that adding the element of vibration reduced measures positively correlated with CS compared to a no-vibration condition.

### 2.3.3. Visual Blurring

Budhiraja et al. [17] used a technique that blurred the screen whenever a rotation occurred. This was found to reduce CS levels and delay the onset of symptoms. Interestingly, those particularly susceptible to CS benefited the most. Hussain et al. [18] also tried a technique based on the concept of foveated rendering that used the depth of field captured by an eye-tracking headset to blur different display regions. A study they conducted found that SSQ scores were reduced by approximately 66%.

### 2.4. Measures of Cybersickness

In the literature, different subjective and objective measures have been used to identify the level of CS or SS.

### 2.4.1. Subjective Measures

The SSQ has been frequently used in literature to measure the effects of CS. This questionnaire was based on the Pensacola Motion Sickness Questionnaire (MSQ). There were found to be enough differences between MS and SS to motivate the creation of the SSQ. The questionnaire is administered post-exposure and aims to collect information on 16 symptoms, each of which is answered with a four-point Likert scale. Symptoms are split into the components of disorientation, nausea and oculomotor issues. These can be summarised with a total severity index [33].

### 2.4.2. Physiological Measures

CS and SS have been associated with changes in Heart Rate (HR), Galvanic Skin Response (GSR), skin conductance, Electroencephalography (EEG), postural stability, electrogastrography (EGG), eye-tracking data, voluntary duration of experience and reaction time [13,34,35] (2017 Oculus Best Practice Guide).

HR is regulated by the sympathetic and parasympathetic systems of the autonomic nervous system. CS has been found to have a positive correlation with HR [34]. It has also been found to be elevated in subjects experiencing pronounced nausea [36]. Additionally, strenuous exercise, fear, stress and anxiety are common factors that influence HR (Tachycardia: Causes, Types, and Symptoms: https://www.webmd.com/heart-disease/atrial-fibrillation/what-are-the-types-of-tachycardia (accessed on 15 March 2023)).

While an Electrocardiogram (ECG) is ideal for accurately monitoring and recording HR, it is not always practical. It requires at least three bioelectrodes and can greatly restrict a person's flexibility of motion [37]. Alternatively, Photoplethysmography (PPG) is an uncomplicated and inexpensive option [37] which provides information on the frequency at which blood is pumped [38]. As for accuracy, it was found that the accuracy and reliability of these devices decrease with increased hand movements. Additionally, Bent et al. [39] found from testing a range of PPG sensors that they were good for resting or prolonged elevated HR, but not so much for responding to changes in activity.

In summary, there does not seem to be a perfect physiological measure that can reliably indicate the presence of CS symptoms in VR, given the normally active nature of interactions in VR (e.g., rapid arm and body movements). However, there are some that could give an indication of what users are experiencing.

### 2.5. Summary

Two key CS theories are the sensory conflict hypothesis and the postural instability hypothesis. These attribute CS and closely related sicknesses to the body rejecting inconsistencies between senses and the body's failure to adapt to conflicts or new dynamics in an experience. A range of factors and strategies for mitigating CS have also been

identified. These primarily relate to the visual display, control, virtual movement and multi-sensory elements.

CS reduction techniques have been used to alter the visual experience of users such as reducing FOV (statically and dynamically) under certain conditions [12,16] and blurring techniques [17,18]. These were successful in reducing the level of CS, except when FOV was manipulated while using amplified head rotations as a control input [16]. Factors such as flicker and speed also have been identified as having a role in CS. However, manipulations that constrain the visual experience can compromise how convincing and engaging it is. It would, therefore, be useful to discover more ways of reducing CS with minimal disruption to the experience, such as the depth-of-field blurring technique used by Hussain et al. [18].

The research by Debarba et al. [19] on changing the level of embodiment in a VE found no significant changes in CS. This was likely due to the lack of CS inducement from not moving in the VE. Interestingly, it has been suggested that embodiment could affect postural stability [1,19,20]. It would, therefore, be useful to test body ownership with member representation in a highly CS-inducing experience with many sensory conflicts, significant optical flow and variation in movement.

Jung et al. [21] conducted research using floor vibration to resolve a visual-vestibular sensory conflict. However, factors such as how well a simulated sense is matched, the number of senses simulated and how similar it is to real experiences people are familiar with could have an impact on the level of CS, especially for simulations with significant movement and variations in altitude within a VE. With many applications that do not recreate real experiences, understanding this effect could be important.

## 3. Material and Methods

### 3.1. Hypotheses

This research aims to identify and understand strategies that can mitigate CS. In the background section, a range of techniques were explored which have been employed. However, in many cases, there is a trade-off and limited suitability for all applications. Considering what has been done successfully and what strategies are yet to be tried to mitigate CS, the following hypotheses were formulated to form the basis of an experiment:

- Hypotheses 1: Manipulating FOV in VR with a vignette based on virtual speed and eye-gaze direction mitigates CS.
- Hypotheses 2: Adding a first-person perspective with member representation to a VR experience mitigates CS.
- Hypotheses 3: Simulating vibration and airflow in a VR experience mitigates CS.

### 3.2. Experience Design

A baseline physical VR setup was designed to provide users with the agency of movement in a 3D environment. It was also built to induce CS symptoms in the majority of users. This involved designing a physical travel interface to control the locomotion of its virtual counterpart through a VE (Video Demonstration of the Experience: https://www.odt.co.nz/star-news/star-christchurch/two-decades-virtual-reality-tech-seen-christchurch-lab (accessed on 21 April 2023)) (Figure 1). To test each hypothesis, different experimental conditions were applied on top of this.

### 3.3. Travel Interface

The flying broomstick travel interface we created (Figure 2), inspired by the work of Beckhaus et al. [40], involved a Swopper stool by Aeris (Aeris Swopper Stool: https://en.aeris.de/products/aeris-swopper (accessed on 15 March 2023)) which could swivel 360°, tilt in any direction and had an adjustable height. Creating this interface involved replacing the seat from the stool with a wooden shaft. A bracket of mild steel was made to affix the shaft onto the stool by bolting its top to the shaft and wedging the stem of the stool into the hollow pipe at its bottom. A seat was also added, attached with screws and glue. Above this, leather padding was placed, connected with VELCRO strips. Users were,

therefore, able to sit astride the travel interface, holding the shaft in front, and manoeuvre it into different orientations by leaning and leveraging it around with their feet. To prevent the stool from toppling over, its base was fastened to a square of MDF to brace against moments. Bolts and metal strips were used to anchor the stool at four different points which could be tightened from above.

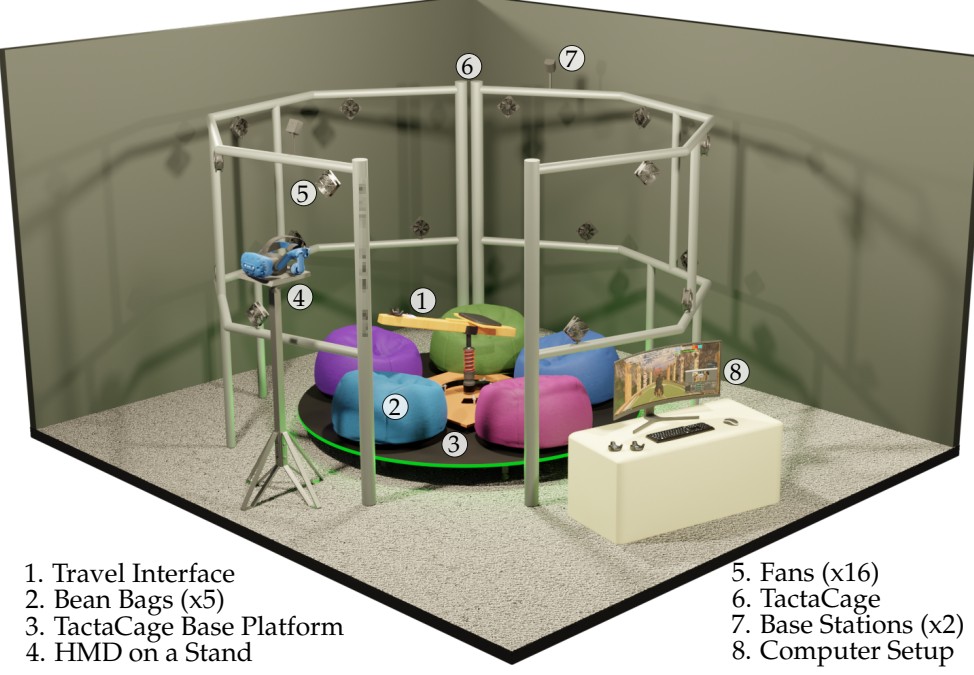

1. Travel Interface
2. Bean Bags (x5)
3. TactaCage Base Platform
4. HMD on a Stand

5. Fans (x16)
6. TactaCage
7. Base Stations (x2)
8. Computer Setup

**Figure 1.** The physical setup of the flying broomstick experience.

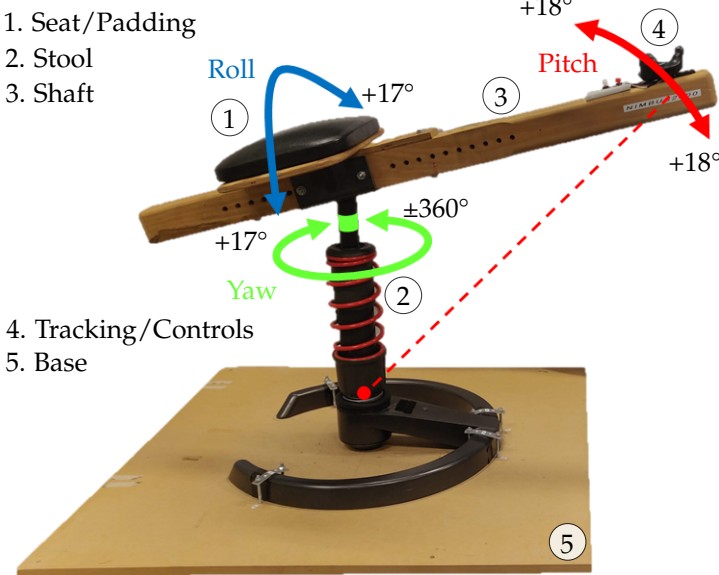

1. Seat/Padding
2. Stool
3. Shaft

4. Tracking/Controls
5. Base

**Figure 2.** The flying broomstick travel interface, its degrees of freedom and the approximate movement range for a user.

### 3.4. Control System

A four Degree-of-Freedom (DOF) control system was designed, capable of capturing the orientation of the interface shaft and the input of two buttons using a Vive tracking puck (Figure 3). This was placed at the end of the shaft with a 3D printed mount (Vive tracker pin connector convertible base: https://www.thingiverse.com/thing:2211803 (accessed on

15 March 2023)) and was able to capture the roll, pitch and yaw of the travel interface. With this information, the orientation of the virtual representation (Figure 4) was set.

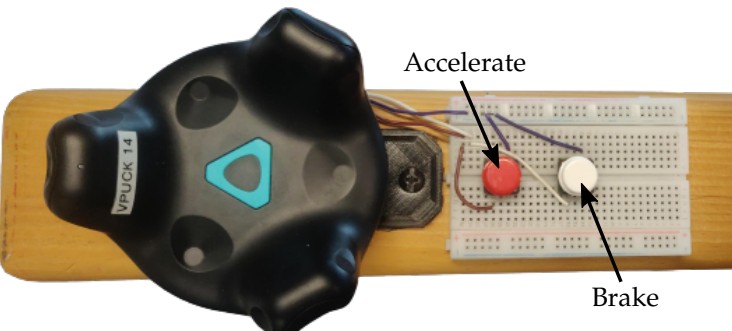

**Figure 3.** The orientation tracking puck and movement controller.

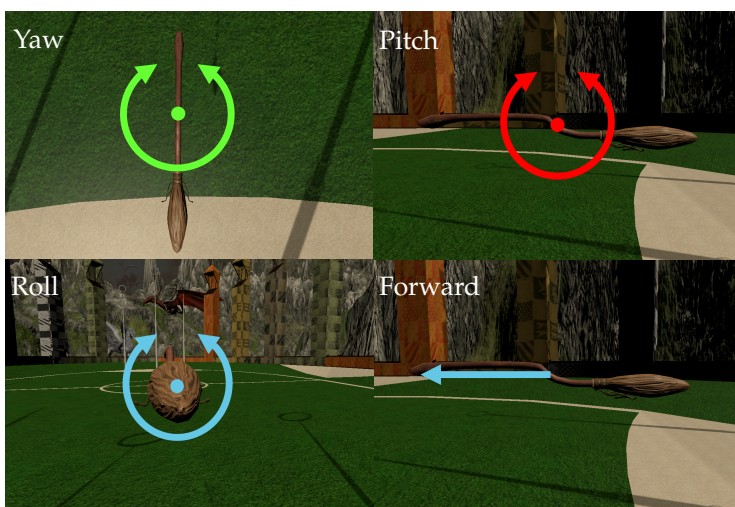

**Figure 4.** The controllable DOF in the VE.

Unlike the physical interface, the virtual counterpart was able to leave the ground and translate. This was controlled by two buttons wired to digital input pins on the Vive tracker (Figure 3). One was for accelerating in the direction that the shaft pointed and the other for braking. These functions were applied by holding the buttons down, with the maximum speed capped at $200 \text{ ms}^{-1}$ and a minimum speed of $0 \text{ ms}^{-1}$. Due to the limitation of the real interface to pitch forward and back, vertical components of the virtual movement were incorporated based on upper and lower pitch thresholds of the interface to enable the user to move up or down more rapidly. During motion gravity was enabled, allowing the user to go faster downwards than upwards. However, with no motion, this was disabled so the user could virtually hover. To simulate turbulence from air currents and induce higher levels of CS, the virtual counterpart was set to move slightly in random directions. These movements were set to be more intense at greater heights, but less so with more speed.

### 3.5. Virtual Environment

To enable users to fly around by interacting with the travel interface, a VE was created in Unity3D for VR (Figures 5 and 6). This was designed to be engaging with a clear objective that guided users through the level, but also provided them with the freedom to roam within certain bounds. The map consisted of a finite rugged terrain with many high peaks and sharp inclines, and was moulded into a basin with a ceiling to keep users on the map. Arranged in a loop, 27 hoops were placed on the map for users to fly through. However, if a user hit a rim of a hoop, they would collide. To provide help with navigation between hoops, the hoops were also set to periodically pulsate red. Various 3D models were also added to enhance enjoyment and evoke nostalgia.

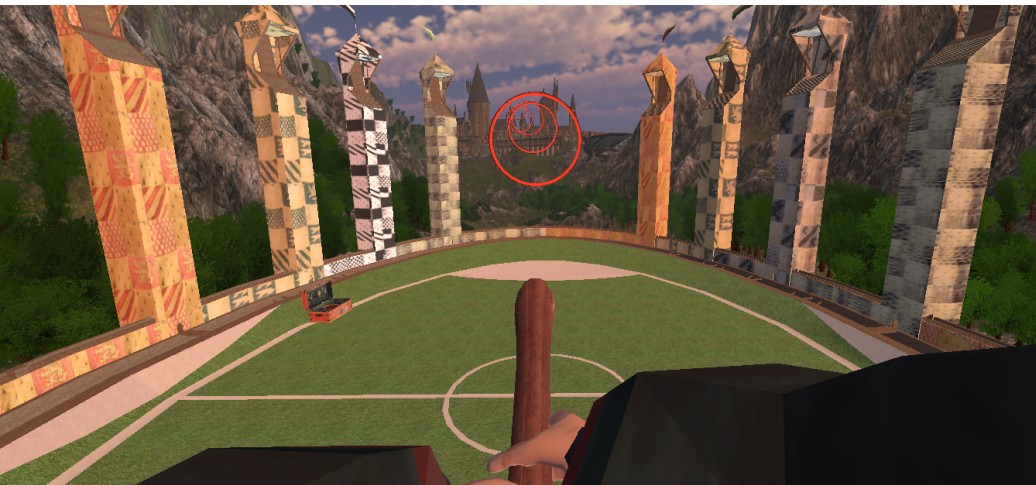

**Figure 5.** The beginning of the course from the perspective of the user.

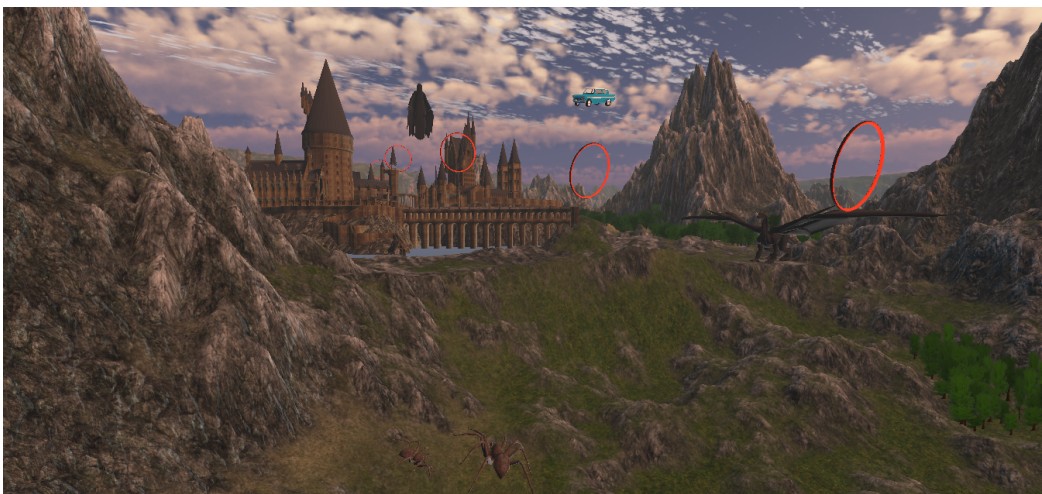

**Figure 6.** The terrain and models featuring in the VE.

The point of view of the user in the VE was consistent with their position on the broomstick interface, allowing them to look around with the VR HMD. Sounds are also featured in this experience. Calming instrumental music played in the background continuously. There were also sound effects that were played at certain events, movements or positions. These included wind at high speeds, collisions, flying through hoops and proximity to certain objects such as a roaring dragon.

*3.6. Experiment Design*

An experiment was designed to test the hypotheses by comparing three manipulation conditions to a control condition (CT). This was done with a between-subject design, exposing each individual to only one of the conditions. Participants who signed up for the study were randomly allocated across four groups, each representing an experimental condition. To ascertain the extent of CS experienced by participants, a range of subjective and objective measurements were taken. An outline of the experimentation process is shown in Figure 7. Approval for this experiment was also granted by the University Human Research Ethics Committee (HREC).

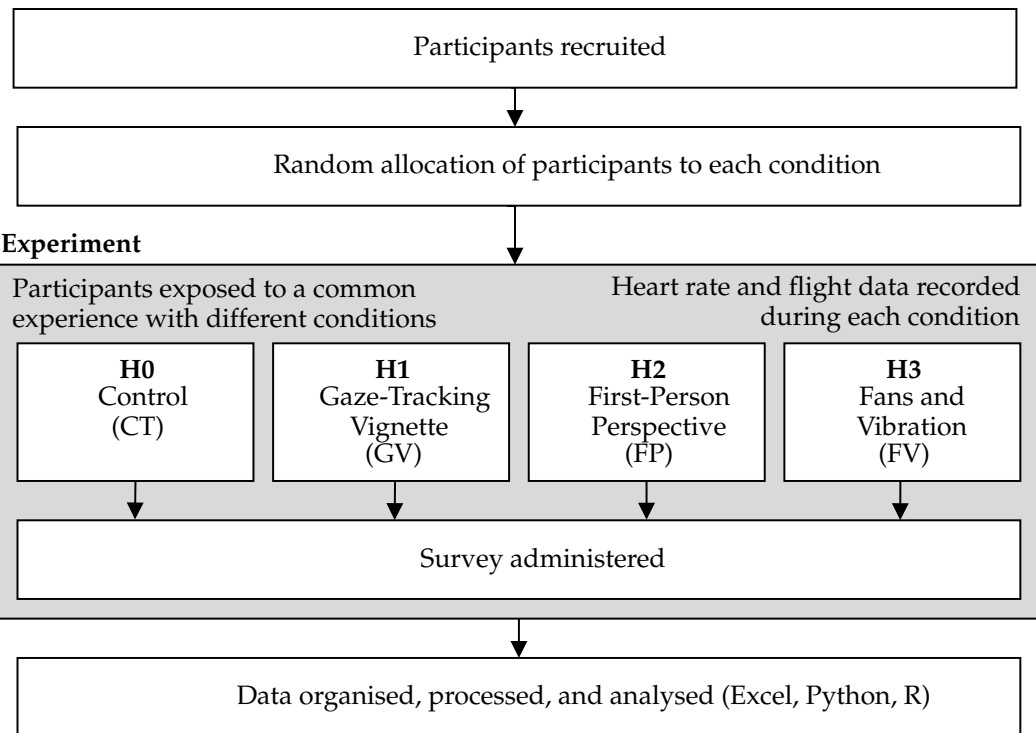

**Figure 7.** An overview of the stages and allocations to conditions involved in the experiment.

### 3.6.1. Conditions

Control. This baseline condition involved the participants wearing the HMD and using the travel interface to control their movement and orientation in the VE. The virtual broomstick was visible in the FOV and the sound was audible through the HMD headphones. The suggested task was to fly through the hoops but was not enforced. A 15-min time limit was also imposed, but participants were able to finish early.

Gaze-Tracking Vignette. A novel dynamic vignette effect was used to manipulate the FOV based on speed and eye-gaze direction as shown in Figure 8. A black vignette effect with an elliptical shape and a smooth gradient was generated using the post-processing effect in Unity3D, with its intensity set to be proportional to the speed of the virtual user. This was configured to be unobtrusive at the maximum possible speed. However, unlike previous vignetting approaches that centre the vignette on the frame [12,16], we used the eye-gaze direction as the central position of the vignette, which meant the vignette would follow where the user's eyes were looking, moving vertically and horizontally. This was implemented using an eye-gaze vector from the Tobii XR SDK in Unity3D and a Vive Pro Eye VR headset, requiring calibration for each participant. To improve the experience, an algorithm was implemented to reduce the jitter in the vignette position by averaging the horizontal and vertical coordinates over ten frames. It, therefore, took at most ten frames to update the vignette to the most recent changes in eye-gaze direction.

First-Person Perspective. A rigged model of a body was imported into Unity3D to provide a sense of body ownership. This was adjusted to a sitting position with its knees bent and both hands clasping the shaft of the virtual broomstick. Then the head of the character was masked out to avoid obstructing the camera tracking the VR headset, allowing users to look around and see different limbs.

To provide embodied interaction, hand tracking was also implemented by fastening a Vive tracking puck to each wrist with adjustable straps (Figure 9) and performing inverse kinematics. Based on the position and orientation of each tracker, the entirety of each virtual arm could be moved. An approach was also taken to guide each hand onto the shaft by "snapping" the hands to it within a certain proximity.

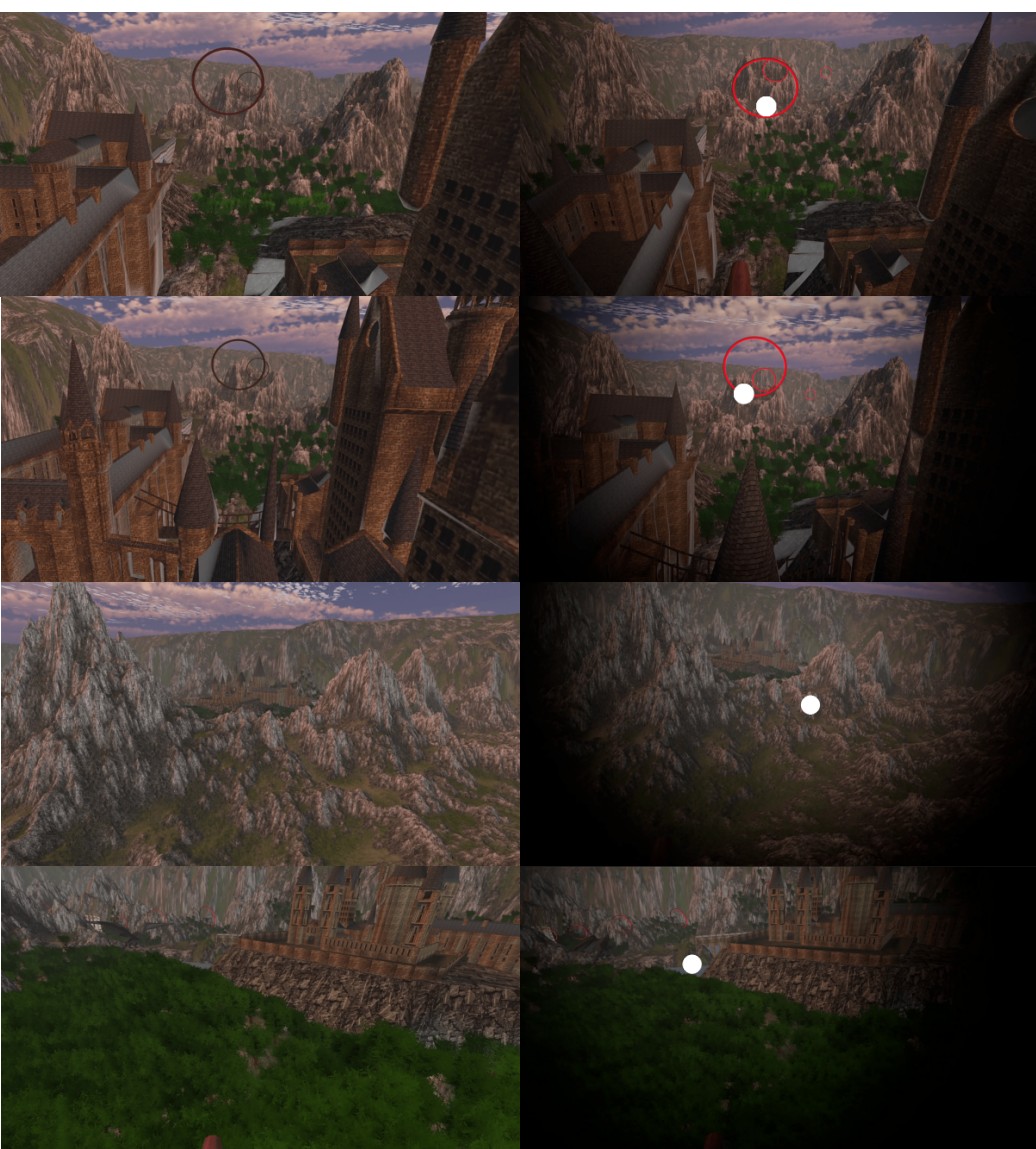

**Figure 8.** A comparison of frames from the point of view of the user between the control (**left**) and GV (**right**) conditions for different speeds and eye-gaze directions. White dots (not visible in the VE) have been added here for clarity to indicate the centre of each vignette.

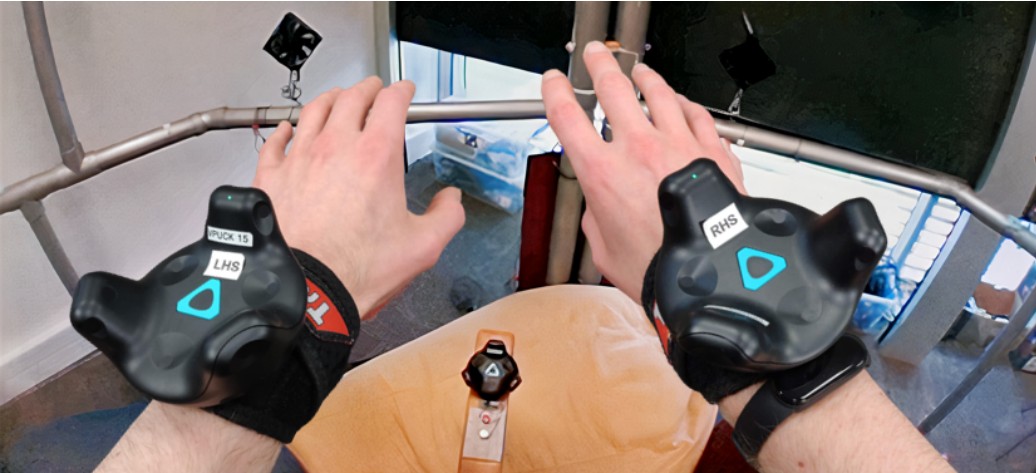

**Figure 9.** An example of how Vive tracker pucks were strapped to each hand.

Examples of this condition are shown in Figures 10 and 11. The frames show a variety of 2D views of FP from the point of view of the user at different head rotations and hand movements.

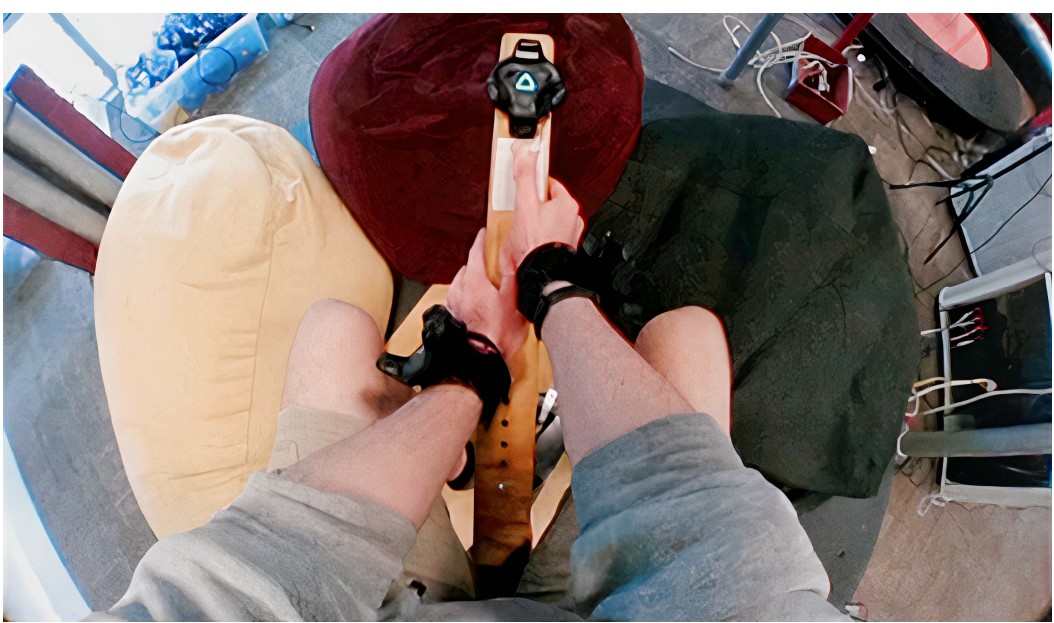

**Figure 10.** An example of gripping the broomstick interface shaft.

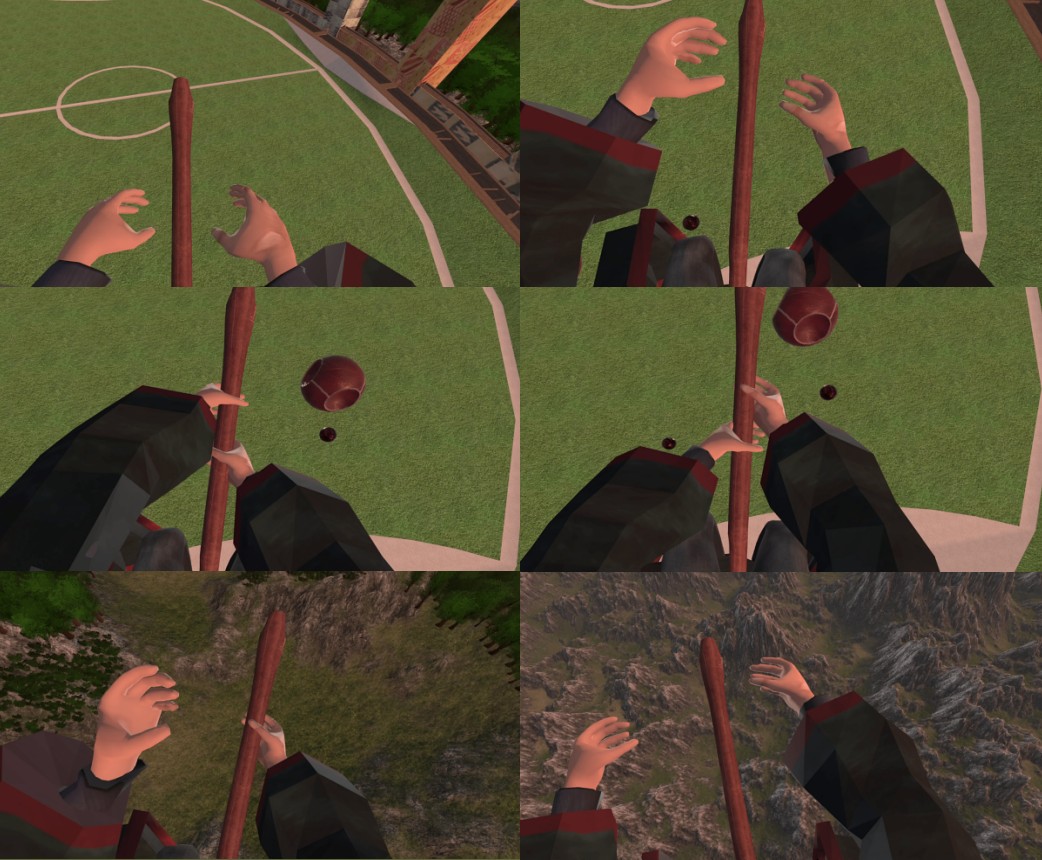

**Figure 11.** Frames from the point of view of the user showing FP at different head rotations and hand movements.

Fans and Vibration. To stimulate different senses and mimic the experience of actual flight, a custom-fabricated enclosure called a TactaCage [41] (Figure 1) was used. This

consisted of fans arranged on an octagonal frame and a raised concentric platform with four haptic transducers (Buttkicker LFEs) mounted underneath (Figure 12), operated with a 1000 W amplifier [21].

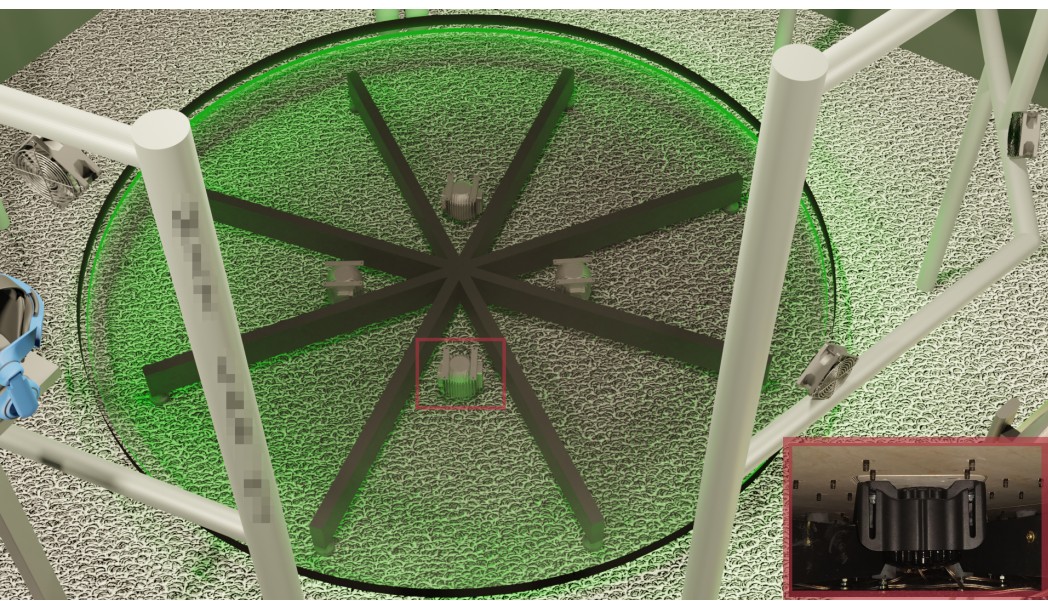

**Figure 12.** Four Buttkicker haptic transducers underneath the TactaCage platform.

The transducers provided the sensation of vibration. While these used the same audio channel as the headphones in the HMD, only low frequencies were transmitted. Therefore, to emphasise the lower frequencies of sounds, they were isolated with a low-pass filter and amplified. This was used to augment sound effects such as collisions and airflow proportional in volume to movement speed.

On each of the two tiers of the TactaCage (Figure 1), eight computer fans were arranged on each side of the octagon facing towards the centre. The fans served to simulate the sensation of local airflow that is experienced when moving through the air at high speed. This involved tracking the rotation of the user and activating the corresponding fans in front of them. These were activated at full intensity during any movement through the VE. There was also a low-intensity global airflow that simulated wind coming from one direction irrespective of user movement and rotation.

### 3.6.2. User Interface

For this experiment, it was important to be able to set up different conditions for each participant, ensure that data was successfully being collected, monitor the VE and monitor what was happening in the real world. To satisfy these requirements and simplify the experiment, a dashboard overlay on the VR application was created that only showed on the computer (Figure 13). This allowed the researcher to switch between conditions, enter a participant identifier for file naming, control the simulation state and activate eye-tracking calibration. It also displayed temporal readouts, user speed and a count of the hoops flown through.

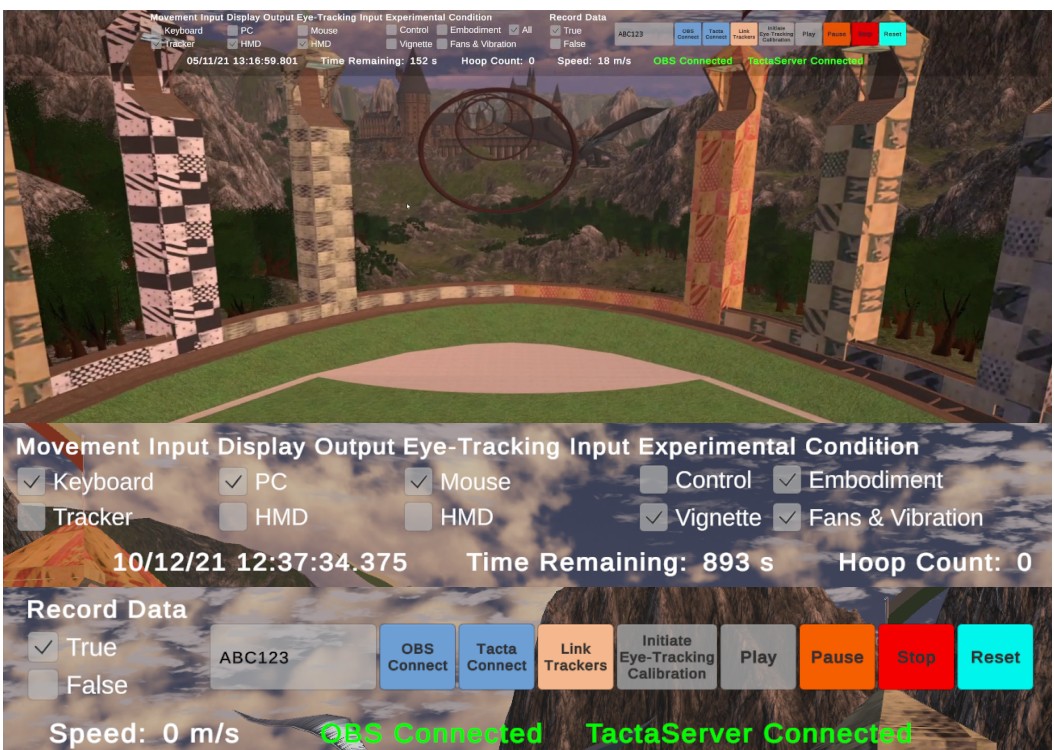

**Figure 13.** The user interface of the VR application for configuring, operating and monitoring the experience.

### 3.6.3. Data Collection

Subjective and objective measurements were taken to gain insight into how each experimental condition affected participants.

Heart Rate Data. A Xiaomi Mi Band 6 Fitness Tracker (Mi Smart Band 6: https://www.mi.com/global/product/mi-smart-band-6/ (accessed on 15 March 2023)) was used to collect an estimate of instantaneous HR with its PPG sensor, recording readings with a frequency of 2 Hz. This was given to participants to put on the wrists of their non-dominant hands. If they were doing the FP condition, the band was placed behind the Vive tracker strap. To record data, the band was connected to an Android phone via Bluetooth and an app called StraMi was used. As soon as the participant put the band on before the experience, the recording was started. Data readings were measured in BPM to the nearest integer and were logged approximately every two seconds. The data were recorded in comma-separated form, with each line containing a timestamp and an HR value.

Flight Data. Time series data were collected on the user's flight through the VE at approximately 90 Hz. This was done automatically by the VR application during the experience. The data included the virtual position and orientation of the user, the number of hoops flown through, speed, distance travelled and height above the ground. This data was saved in separate CSV files for each participant.

Visual Recording. Participants were recorded from three different angles using web cameras (720 p Microsoft LifeCam HD-3000). These consisted of two views from the side and an aerial view from the ceiling. The maximised VR application window was also recorded, showing the point of view of the user. Open Broadcaster Software (OBS) was used to record the camera views and the application into a single feed. This feed was captured at 30 FPS with a resolution of 1080 p. To automate the capture in OBS, the obs-websocket add-on was used to run a WebSocket server. This allowed the Unity3D application to form a connection with OBS, trigger recording and set file names to the user input once the simulation started.

Survey Data. Post-exposure, a survey was administered using the online Qualtrics platform and required users to select options on Likert scales and write into text entry fields. The first part of this form was used to gather background information on participants including age, gender, and experience with VR (a rating out of four). They were also asked to describe their subjective experience in a text field with question prompts including "What words would you use to describe this experience?", "How did the experience make you feel?" and "Did you notice anything distracting or annoying? If so, what?". In the second part, the SSQ was used to assess the level of SS they experienced based on their rating of certain symptoms.

### 3.6.4. Data Processing

Due to the discontinuous nature of data recorded for the distance travelled, it could not be numerically differentiated reliably. To resolve this, it was modelled for each data set using splines with 200 knots. This resulted in smooth lines that were able to be differentiated to speed and acceleration [42]. The SSQ survey results were also used to calculate the component scores based on the weightings of each symptom and a total severity score using the method described by Kennedy et al. [33].

### 3.6.5. Data Analysis

To gain insight into the population sample, the survey responses for the age, gender and experience with VR of participants were collected and plotted as histograms.

To inspect the SSQ results, a box plot for the total severity comparing each condition was generated. To check for evidence of significant differences in the means of SSQ scores between conditions, a one-way ANOVA test was chosen with the following hypotheses:

$$H_0: \quad \mu_{ct} = \mu_{gv} = \mu_{fp} = \mu_{fv}$$
$$H_a: \quad \text{The means are not all equal.}$$

In this test and the subsequent tests, a significance level of 10% ($\alpha = 0.1$) was assigned, as opposed to the standard 5%. This was chosen due to the time and recruitment limitations on getting enough evidence which meant that this testing was more aimed at a preliminary indication of how the strategies affect CS.

To assess the validity of doing an ANOVA, a Shapiro-Wilk test was used to check the normality of the distributions ($H_0$: Normal, $H_a$: Non-normal), followed by Levene's test to check the homogeneity of variances ($H_0$: Homogeneous, $H_a$: Non-homogeneous). To maximise the success of these tests, a square transformation followed by a cube-root transformation was applied to the data. An outlier result of one participant that did GV was also removed due to being significantly higher than any other result. To investigate each treatment group against CT, a post-hoc test was conducted using Dunnett's Method with the following hypotheses:

$$H_0: \quad \mu_{ct} = \mu_{gv} \quad \mu_{ct} = \mu_{fp} \quad \mu_{ct} = \mu_{fv}$$
$$H_a: \quad \mu_{ct} \neq \mu_{gv} \quad \mu_{ct} \neq \mu_{fp} \quad \mu_{ct} \neq \mu_{fv}$$

This was followed by a 90% confidence interval to determine whether significant differences were increases or decreases. The effect size (Cohen's f statistic) was also calculated in G*Power [43] using the means of each condition and a pooled standard deviation [44].

To understand whether gender, age and VR experience have an impact on CS in this study, the correlation (Pearson correlation coefficient) of each against total severity scores from the SSQ was calculated. This also involved testing for the significance of the correlations ($H_0$: True correlation is 0, $H_a$: True correlation is not 0). For the calculation of gender, this variable was assigned a 1 for Females and a 2 for Males (point-biserial correlation).

The HR variation across all participants was summarised by splitting the data into three stages. These were baseline HR values recorded immediately before the experiment

started, the shift from the mean baseline for each participant recorded in the first half of the experiment (0–7.5 min) and then the shift from the mean baseline recorded in the second half (7.5–15 min). The variation was reported for each condition with minimum HR, maximum HR, mean HR and standard deviation HR.

To analyse the factors that influence change in HR relative to the mean HR prior to each experiment, multiple linear regression models with interactions were used to determine whether the relationships are positive or negative. The Movement Variables (MVs) in these models included the distance travelled, speed, acceleration, deceleration and height above the ground. Each model was created in terms of one of these MVs and categorical variables for each manipulation condition applied on top of CT. The interactions arose from the interplay between each condition and the MVs having an impact on HR. The general formula for each model was:

$$
\begin{aligned}
y_{hr} = \beta_0 &+ \beta_{mv} x_{mv} + \beta_{gv} x_{gv} + \beta_{fp} x_{fp} + \beta_{fv} x_{fv} \\
&+ \beta_{mv:gv} x_{mv} x_{gv} + \beta_{mv:fp} x_{mv} x_{fp} + \beta_{mv:fv} x_{mv} x_{fv} \quad (1)
\end{aligned}
$$

where:

$$
\begin{aligned}
y_{hr} &= \text{HR} \\
\beta_0 &= \text{constant term} \\
\beta_{mv} &= \text{MV coefficient} \\
x_{mv} &= \text{MV} \\
\beta_{gv}, \beta_{fp}, \beta_{fv} &= \text{condition variable} \\
&\quad \text{coefficients} \\
x_{gv}, x_{fp}, x_{fv} &= \text{condition variables} \\
&\quad \text{(1 or 0 based on which} \\
&\quad \text{condition is enabled)} \\
\beta_{mv:gv}, \beta_{mv:fp}, \beta_{mv:fv} &= \text{interaction coefficients} \\
&\quad \text{between MVs and} \\
&\quad \text{conditions}
\end{aligned}
$$

Hypothesis tests for the coefficients of each model were also conducted to determine if they were significant ($H_0$: $\beta = 0$, $H_a$: $\beta \neq 0$). If a $p$-value was above the 0.1 threshold, the coefficient was rejected from the model due to a lack of evidence that it was not zero. To assess the validity of the models, each was checked for linearity, independence, homoscedasticity and normality of residuals. Using residual plots and quantile-quantile plots, each model was found to satisfactorily meet these assumptions.

Observations from the researcher and survey feedback from users on the experience were also collated. These were summarised to include trends and notable results.

### 3.7. Equipment and Software

### 3.7.1. Computer Hardware

The computer setup used in this experiment was custom-built in the lab with the following specifications:

- Processor: Intel Core i7-8700 (3.20–4.20 GHz)
- Baseboard: Gigabyte Technology B360 Aorus
- Physical memory: 32.0 GB RAM
- Graphics card: NVIDIA GeForce RTX 2080
- SSD (OS and Programs)
- HDD (Unity Application and Data)

### 3.7.2. Safety

Participants needed to be safe even if they fell off the stool. Therefore, five bean bags were arranged around the interface (Figure 1). Cabling extensions were also added to enable the headset to be routed via the ceiling and avoid tangling around participants on rotation.

### 3.7.3. Application Development

A range of software packages, tools and assets was used in the development of the VR application. These included Unity3D (2020.3.20f1) from Unity Technologies, SRanipal Runtime (1.3.1.1) by Vive, SteamVR (1.18.6) by the Valve Corporation and OBS Studio (27.1.3) which is an open-source project.

### *3.8. Participants*

People with vestibular, balance, dizziness and migraine issues were identified as being a demographic susceptible to extreme CS and, therefore, high levels of discomfort. This was not seen as appropriate for this experiment. Therefore, those who knew they had any of these issues and followed the screening advice did not participate in the study. A mass restriction was also set to 100 kg to protect the equipment, but was not enforced strictly. Participants were additionally required to be at least 18 years old.

A variety of advertisement methods were employed to recruit participants for this experiment. This involved posting on Facebook, posters and sending messages via email lists. Potential participants were also made aware of the study by word of mouth. To entice people to sign up and do the experiment a \$15 gift card was offered as an incentive.

Prospective participants were directed to a web page created by the booking platform https://bookwhen.com (accessed on 15 March 2023) where they could sign up for the experiment. This page had a list of available 45-min time slots and had details on the experiment.

### *3.9. Experimental Procedure*

For each experiment, 45 min were allocated, with 15 min between each subsequent subject. Prior to each session, users were instructed on how to use the travel interface and what to do in an information sheet they were given to read. This was then reiterated by the researcher verbally. Immediately after participants were on the interface with the HMD on, the eye-tracking and IPD were calibrated and the seat height was adjusted if required. This provided participants with a brief period of time to familiarise themselves with the physical interface and the VE where they were allowed to rotate, but not translate until the experiment was started by the researcher. Once started, the data collection and the user were monitored. If there were any issues such as the bean bags or cabling restricting movement, adjustments were made. After 15 min of exposure elapsed, questions were asked to participants on how they were feeling and in some instances behaviours or symptoms that the researcher noticed. Participants were then directed to a computer to fill out the questionnaire.

## 4. Results

### *4.1. Participant Information*

Thirty-seven people participated in this study across four conditions. This comprised 10 allocated to CT, nine to GV, nine to FP, and nine to FV. The age, gender and experience with VR distributions are shown in Figure 14, Figure 15 and Figure 16, respectively. Due to the COVID-19 environment, participant recruitment was challenging. There were also instances where participants did not show up to their allocated time slot. These factors led to fewer participants doing the study than intended. A consequence of this was that the random assignment to conditions led to an uneven spread of age, gender and experience.

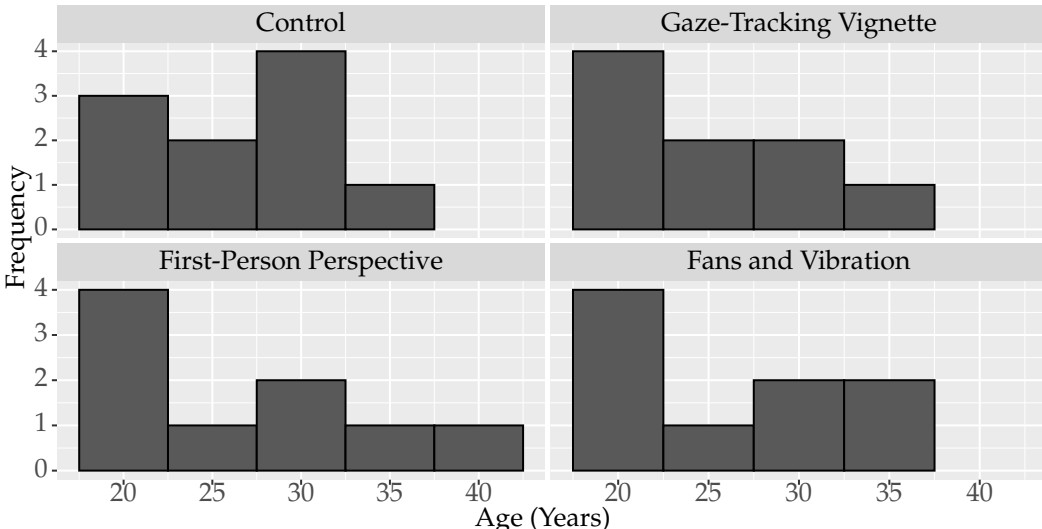

**Figure 14.** The distribution of ages. M = 26 (SD = 5.61).

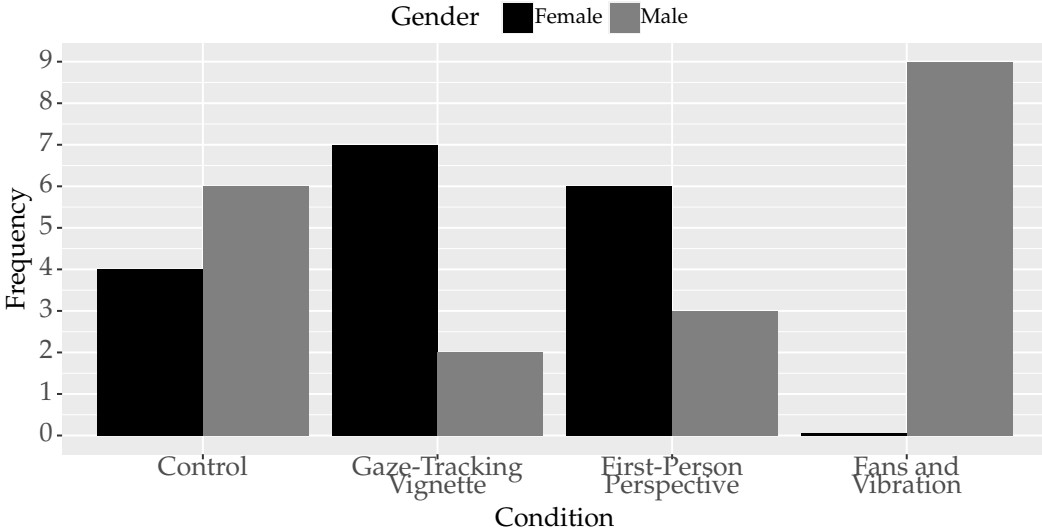

**Figure 15.** The proportions of genders from a total sample of 17 females and 20 males.

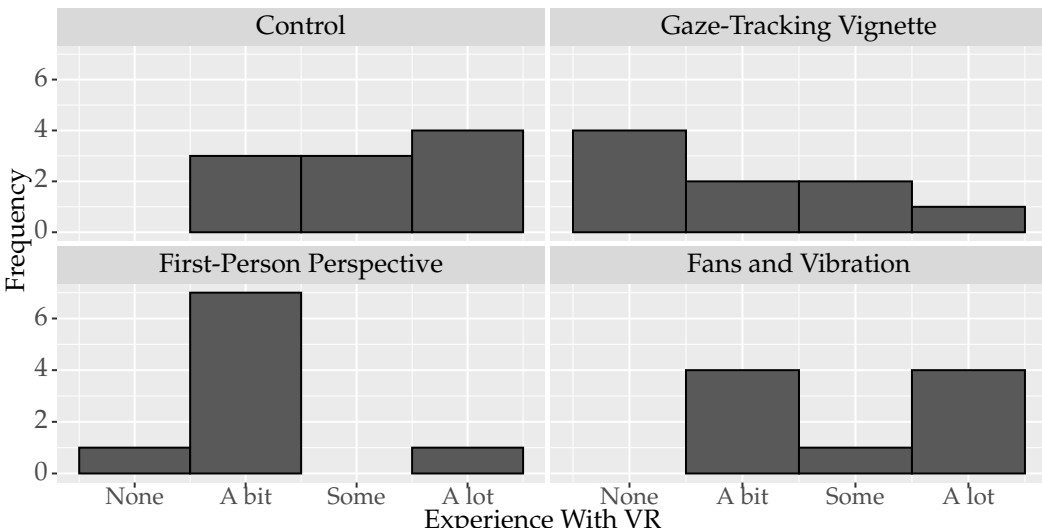

**Figure 16.** The distributions of participants' ratings of experience with VR. M = 2.57 (SD = 1.04).

### 4.2. Simulator Sickness Questionnaire

The SSQ results for total severity have been split into the experimental conditions for comparison as shown in Figure 17. For these results, higher scores indicate higher levels of experienced SS.

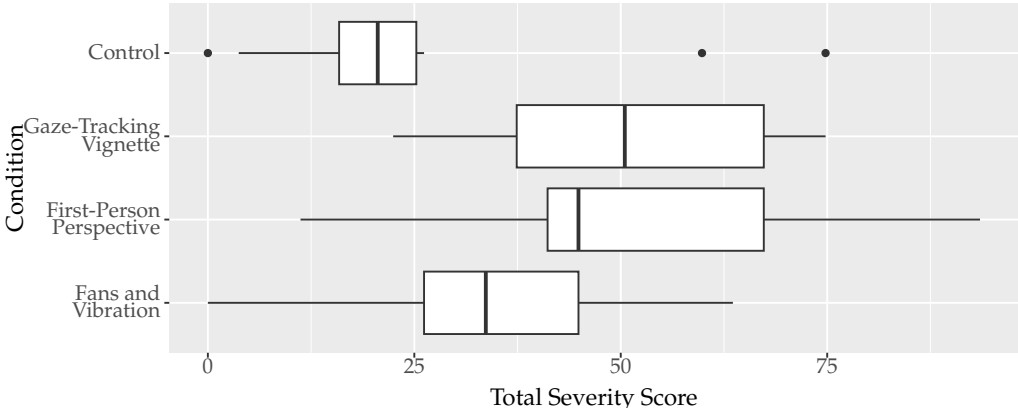

**Figure 17.** A comparison between conditions for the SSQ total severity scores. The maximum range of possible scores is from 0 to 235.62.

#### 4.2.1. Analysis of Variance

The results of testing to see if an ANOVA could be performed are shown in Table 1. The Shapiro–Wilk test for normality provided significant evidence that only the total severity component had conditions that were all normally distributed. Additionally, Levene's test provided significant evidence that the variances of all the data sets were equal. In the subsequent ANOVA (Table 2), significant evidence of a difference in the means of the conditions for each SSQ component apart from nausea was found.

**Table 1.** The results of a Shapiro–Wilk test and Levene's test.

| SSQ Component | Condition | Shapiro–Wilk Test (*p*-Value) | Levene's Test (*p*-Value) |
|---|---|---|---|
| Nausea | CT | 0.4800 | |
| | GV | 0.4340 | 0.4267 |
| | FP | 0.3190 | |
| | FV | **0.0952** | |
| Oculomotor | CT | 0.4490 | |
| | GV | 0.6690 | 0.6372 |
| | FP | 0.5190 | |
| | FV | **0.0055** | |
| Disorientation | CT | **0.0715** | |
| | GV | 0.1650 | 0.8030 |
| | FP | 0.9270 | |
| | FV | 0.6430 | |
| Total Severity | CT | 0.3580 | |
| | GV | 0.3040 | 0.9311 |
| | FP | 0.8210 | |
| | FV | 0.2700 | |

**Table 2.** The results of a one-way ANOVA.

| SSQ Component | Critical Value | F Ratio | *p*-Value |
|---|---|---|---|
| Nausea | 2.2635 | 2.1270 | 0.1160 |
| Oculomotor | 2.2635 | 2.7050 | **0.0618** |
| Disorientation | 2.2635 | 2.8060 | **0.0554** |
| Total Severity | 2.2635 | 2.9360 | **0.0481** |

### 4.2.2. Post-Hoc Analysis

The results of a post-hoc Dunnett's test are shown in Table 3, comparing each manipulation condition to CT. This found significant evidence for differences due to FP for nausea, disorientation and total severity. In addition, significant evidence for differences due to GV was found for oculomotor, disorientation and total severity. However, no evidence was found for significant differences with FV.

**Table 3.** The results of Dunnett's test.

| SSQ Component | Control vs. | Confidence Interval | *p*-Value |
|---|---|---|---|
| Nausea | GV | [−0.6021, 4.4067] | 0.2633 |
| | FP | [0.3175, 5.1692] | **0.0548** |
| | FV | [−1.2835, 3.5682] | 0.6290 |
| Oculomotor | GV | [0.2592, 3.9262] | **0.0519** |
| | FP | [−0.0337, 3.5183] | 0.1086 |
| | FV | [−1.2178, 2.3342] | 0.8443 |
| Disorientation | GV | [0.4569, 6.8004] | **0.0512** |
| | FP | [0.3116, 6.4562] | **0.0630** |
| | FV | [−1.6417, 4.5030] | 0.6367 |
| Total Severity | GV | [0.2161, 4.5678] | **0.0634** |
| | FP | [0.3585, 4.5737] | **0.0454** |
| | FV | [−1.2244, 2.9908] | 0.7045 |

### 4.2.3. Effect Size

The effect size of the four conditions was calculated to be 0.5080 with a pooled standard deviation of 22.0048.

### 4.2.4. Correlations with Participant Information

The correlations (Pearson correlation coefficients) of gender, age and experience using VR with total severity scores calculated from the SSQ are presented in Table 4.

**Table 4.** The correlations of participant information with total severity SSQ scores.

| Variables | Correlation Coefficient | *p*-Value |
|---|---|---|
| Gender vs. Total Severity | −0.0905 | 0.5997 |
| Age vs. Total Severity | −0.3649 | **0.0287** |
| VR Experience vs. Total Severity | −0.3284 | **0.0505** |

### 4.3. Physiological Responses

A summary of the variation HR for different stages of the experiment is shown in Table 5. Additionally, the results of the linear regressions between change in HR and different MVs with interactions for each condition are presented in Figures 18–21.

**Table 5.** The variation in HR at different stages of the experiment.

| Statistic | Condition | Baseline (Before) | Shift (1st Half) | Shift (2nd Half) |
|---|---|---|---|---|
| Min | CT | 57.0000 | −13.1774 | −19.1774 |
| | GV | 63.0000 | −18.6204 | −9.6204 |
| | FP | 68.0000 | −26.3684 | −13.1905 |
| | FV | 63.0000 | −29.9462 | −31.4857 |
| Max | CT | 118.0000 | 30.0735 | 29.0678 |
| | GV | 132.0000 | 34.1279 | 29.1279 |
| | FP | 126.0000 | 31.6316 | 40.6316 |
| | FV | 121.0000 | 26.4107 | 38.0385 |
| Mean | CT | 85.0573 | 6.6017 | 6.3168 |
| | GV | 90.9169 | 8.1231 | 9.2679 |
| | FP | 93.7892 | 6.6104 | 11.8681 |
| | FV | 85.0971 | 3.9319 | 4.5709 |
| SD | CT | 12.1265 | 7.6874 | 7.6331 |
| | GV | 14.5803 | 9.1782 | 6.6824 |
| | FP | 11.1942 | 11.5703 | 11.7055 |
| | FV | 11.8774 | 10.7184 | 12.8420 |

Distance. All non-control conditions have higher HRs above 125 km. FV and FP have lower HRs at 0 km. FP changes the most with the highest gradient, and CT has the gradient closest to zero.

Speed. All non-control conditions have higher HRs at 200 ms$^{-1}$. FP and FV have lower HRs at 0 ms$^{-1}$, but GZ is higher. FP changes the most with the highest gradient and CT has a zero gradient.

Acceleration. GZ has a higher HR at −50 m$^{-2}$. FV has a lower HR at 0 ms$^{-2}$, but with FP and GZ higher. FV has the biggest change while CT has a zero gradient.

Deceleration. GZ has a higher HR at 50 ms$^{-2}$, but with FP and FV lower. GZ and FP have higher HRs at 0 m$^{-2}$, but with FV lower. FV has the biggest change while CT has a zero gradient.

Height Above Ground. GV and FP have higher HRs above 400 m, but with FV lower. GV and FP have higher HRs at 0 m, but with FV lower. FV changes the most, with all other conditions unchanging.

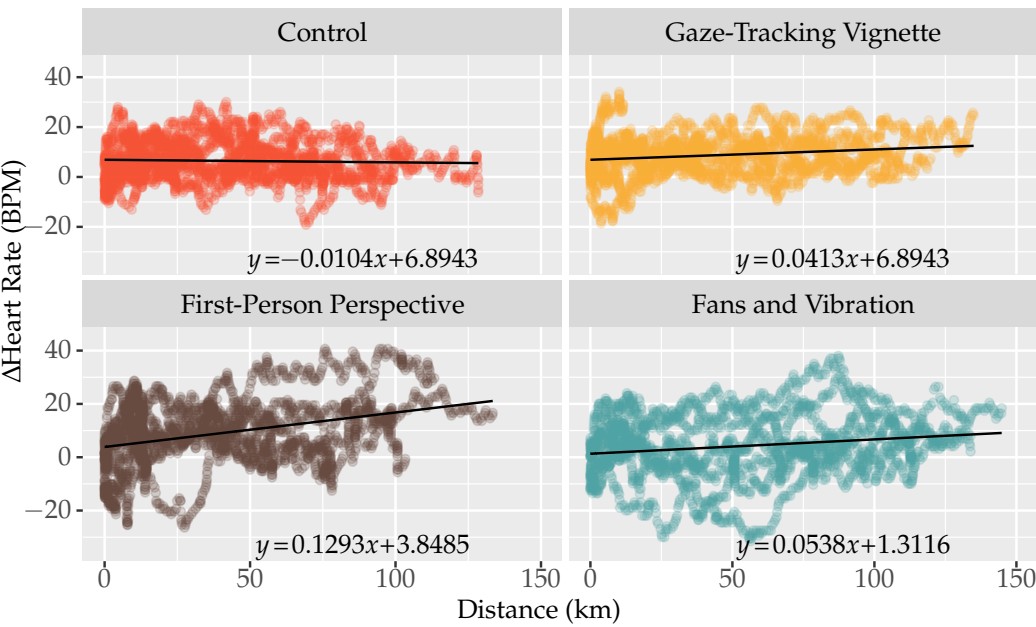

**Figure 18.** A comparison between distance travelled in VR and change in HR relative to a baseline.

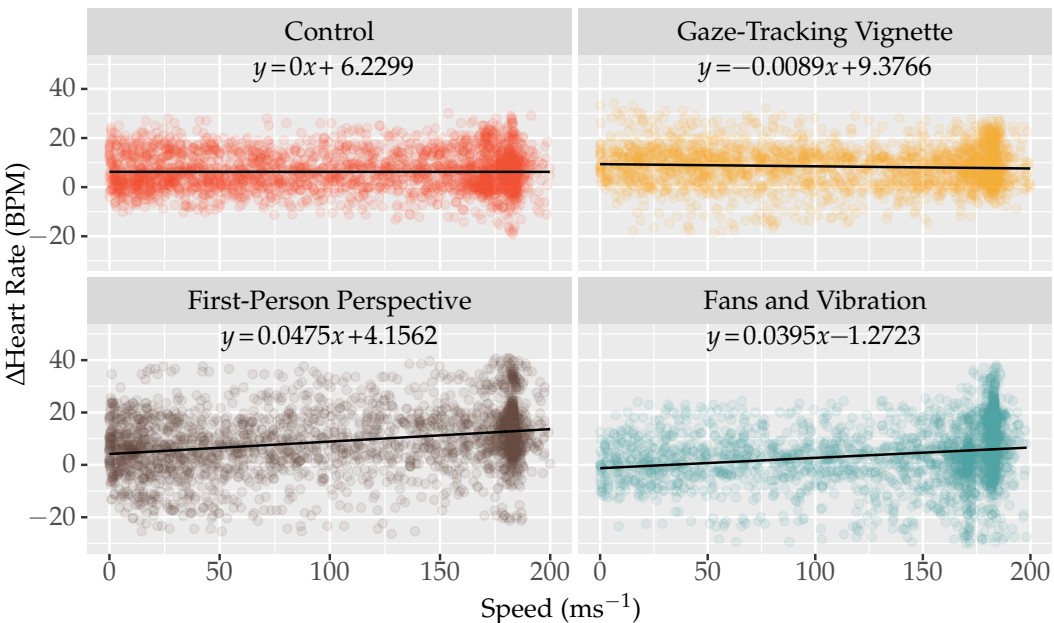

**Figure 19.** A comparison between the speed in VR and change in HR relative to a baseline.

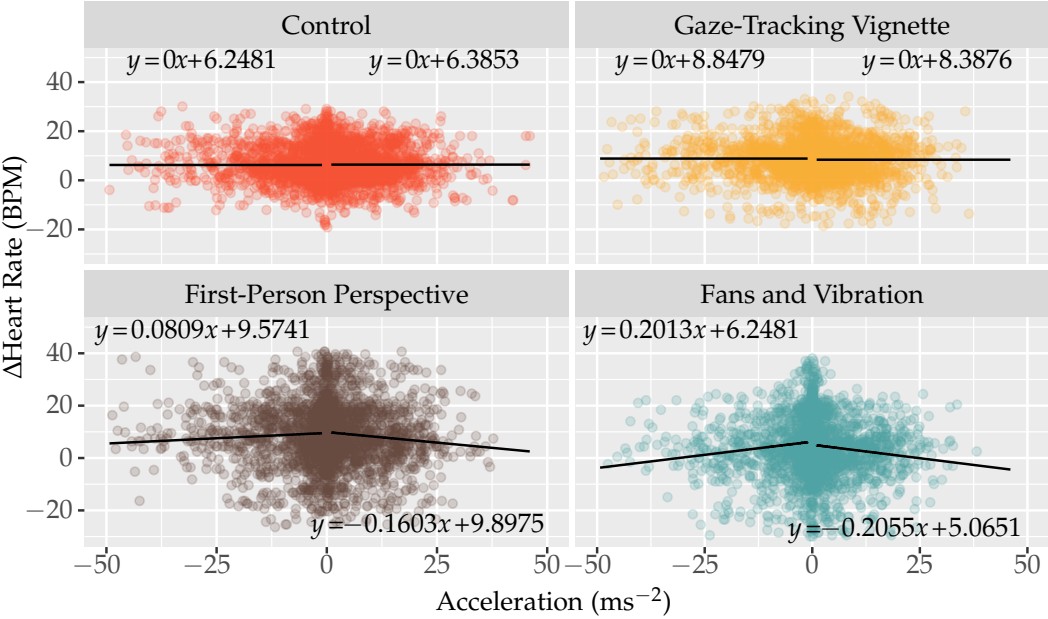

**Figure 20.** A comparison between acceleration in VR and the change in HR relative to a baseline.

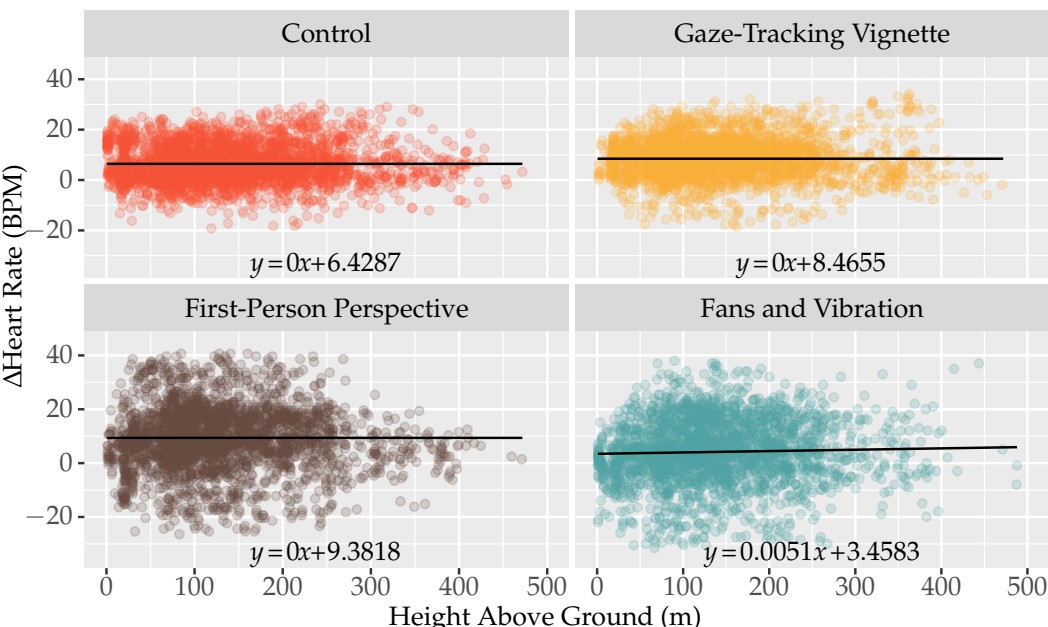

**Figure 21.** A comparison between height above the ground and change in HR relative to a baseline.

*4.4. Observations and Comments*

Overall, most participants found this to be an entertaining and immersive experience. However, some described it as hard, tiring, physically demanding and not providing enough sustained interest.

During the session, some users focused entirely on flying through hoops, while others chose to explore the map after initially flying through some hoops. The brake was rarely used and most users held the acceleration button down to maintain speed, which caused fatigue. Notably, pitching the interface forward to descend for sudden changes in motion was found to be difficult and some found the interface uncomfortable. There were also some issues with obstructions from the bean bags and the cabling, both of which were adjusted as they became apparent. For FP, many appeared to see some of the virtual bodies. However, most only focused on it briefly. Additionally, a few participants noticed that the virtual broomstick bobbed, jittered and rocked from side to side. One participant also noticed the FOV being decreased around the edges for GV.

After the experiment, a small number of people had to remain seated after taking off the HMD to recover due to experiencing symptoms. It was also common to see people slightly unsteady when they stood up and sweating after the experience. This seemed to be in part due to variable heat in the room in which the experiment was conducted. There were many reports of symptoms related to CS. This included feeling ill, sick, dizzy, disorientated, lightheaded, queasy, nauseous and motion sick. Overall, the most common verbally-reported symptom was dizziness. There were also reports of tiredness, pressure from the HMD, fear of heights, stiffness, unsteadiness and blurriness of the display. At the time of finishing each experiment participants did not appear to have lasting symptoms, but data on this and any remaining post-experiment symptoms was not recorded.

## 5. Discussion

*5.1. The Experience*

The most commonly reported symptom related to CS was dizziness. Users also reported increased CS symptoms with increased length of exposure to the experience. This could be due to the cumulative user movement (optical flow), the increasing number of sensory conflicts and the amount of flicker. Moreover, the travel interface did not suit everyone for use for the entire 15 min. It was found to be difficult to tilt to certain angles, unresponsive to button presses and not particularly comfortable. Notably, it was reported

by some that they overheated with physical exertion. Despite this discomfort, some still continued the experiment to the end with the HMD on. While the interface allowed users to have agency over their virtual movements and provided them with the freedom to explore the VE, this "freedom" could have influenced the results by causing frustration and discomfort. It was also observed that the perception of speed and acceleration seemed to change with height. Overall, it was noted that almost all participants enjoyed the entire experience despite some feeling symptoms, having issues with the control or finding it a bit repetitive.

*5.2. Conditions*

The SSQ results showed that across all components each manipulation condition had a higher median score (more SS) than CT (Figure 17). It was also the case that CT always had the smallest IQR. There were also several outliers which, on closer inspection of the data, were in most cases caused by the same group of people across components. These could have introduced some errors in the analysis. Additionally, the effect size calculated is above the threshold of what is considered to be a large effect [44], providing evidence that across the conditions there is some effect occurring.

The results of correlating participant information and SSQ total severity scores (Table 4) provide evidence of significant relationships for age ($p = 0.0287$) and VR experience ($p = 0.0505$). Both of these correlation coefficients are also negative, indicating that with increasing age and VR experience, the SS severity is less. However, the relationship with gender is not significant for this experiment. This result for age is in line with the literature, but not so much for gender [30]. There have also been studies with VR users having lesser symptoms after adapting to experiences [45]. These adaptations have been categorised as behavioural, cognitive and physiological [46]. Over repeated sessions, it has been found that users will develop a tolerance [3,12].

The multiple linear regression models (Figures 18–21) provided insight into how the independent MVs and conditions interacted and contributed to changes in HR. Removing all of the coefficients with evidence to suggest they were zero implied that HR changes due to the independent variables were significant. In the resulting models, it was noticed that all of the CT conditions had positive linear intercepts. There were also positive mean HR shifts from the baseline of each condition Table 5. Moreover, in all conditions apart from CT the HR shift is higher in the second half. This suggests that HR increases irrespective of the condition and in most cases with time. This is not surprising considering the physical manoeuvring and mental workload [47,48] required to control the broomstick interface. Mental workload is also known to affect CS [49,50] which could have played a role in this experiment.

While many factors which can influence HR were not included in these models, and the link between CS and HR is not clearly established in the literature, they are examined together in the following sections.

5.2.1. Control

With increasing travel distance more optical flow, flicker and sensory conflicts would be expected, increasing symptoms of CS. However, the HR gradient for distance travelled was slightly negative, indicating a decrease in HR. This could have been influenced by users becoming more adapted to the experience, or choosing to slow down when they started to feel symptoms. The research indicates that the rate of the onset of SS is proportional to speed [32]. However, the HR gradient for speed in CT was zero, indicating no change in HR. This could have been influenced by the limited range of speed or that speed changes were almost always above the ground where there is typically less optical flow and flicker. Based on the sensory conflict theory, it would be expected that increases in the difference between virtual acceleration and the vestibular system increase CS. However, CT had zero gradients, meaning no change at all with acceleration, indicating no change in HR. This

could have been influenced by the short time that it took to accelerate to full speed or slow down to a stop, especially if it was due to a collision.

At lower heights, more optical flow or flicker would be expected from proximity to objects, increasing the level of CS. Yet, CT had a zero gradient for height above the ground, indicating no change in HR. It is possible users only need to be slightly off the ground for the effect of the optical flow to be minimised.

### 5.2.2. Gaze-Tracking Vignette

The results of conducting Dunnett's test found significant evidence that GV increased the total severity of SS when compared to CT. This was indicated by a *p*-Value of $p = 0.0634$ compared to the significance level $\alpha = 0.1$ (evidence for the rejection of the null hypothesis), and a positive confidence interval of the difference from CT between 0.2161 and 4.5678 (Table 3).

The effect of the GV on HR appeared to be influenced by changes in distance travelled and speed. Compared to CT, GV had a higher HR gradient that was positive for variations in the distance travelled with the same intercept. It also had a lower HR gradient that was slightly negative for variations in speed with a higher intercept. Additionally, there was a constant increase in HR for acceleration, deceleration, and height above the ground directly due to GV. Therefore, each GV result for different MVs had the same or a higher intercept and gradients that were either positive or close to zero. Of all the HR data recorded, the mean HR shift was highest in the first half for GV (Table 5). Overall, these results suggest that GV causes HR to increase.

These results could have been affected by the delay in the vignette introduced by the rolling average of the eye-tracking coordinates or using the incorrect intensity. Due to being out of sync with real-time eye movement, this may have caused users to feel as if their head was moving due to the scope of the FOV change. This kind of involuntary movement could compromise postural stability or cause sensory conflicts leading to worse CS symptoms. However, only one person noticed their FOV decreasing from the vignette, and no one reported noticing their FOV shifting position.

The SSQ provided evidence that GV increased SS and the objective data provided some evidence that GV increased HR with the variation of MVs and over time. Overall, this suggests that the GV condition does not mitigate CS but instead increases it. Therefore, the hypothesis for this condition ($H_1$: Manipulating FOV in VR with a vignette based on speed and eye-gaze direction mitigates CS) cannot be supported.

### 5.2.3. First-Person Perspective

Similar to GV, Dunnett's test found significant evidence that FP increased the total severity of SS compared to CT. This was indicated by a *p*-Value of $p = 0.0454$ (evidence for the rejection of the null hypothesis) and a positive confidence interval of the difference from CT between 0.3585 and 4.5737 (Table 3).

The effect of FP on HR appeared to be influenced by changes in distance travelled, speed, acceleration, and deceleration. Compared to CT, FP had higher HR gradients that were positive for variations in distance travelled and speed with lower intercepts. It also had a lower HR gradient that was negative for variations in acceleration with a higher intercept and a higher HR gradient that was positive for variations in deceleration with a higher intercept. Additionally, there was a constant increase in HR for variations in height above the ground. This suggests that FP could lead to a lower HR for short distances travelled, low speeds or large absolute accelerations. However, with users typically moving at or close to maximum speed, and not accelerating or decelerating for significant periods most of the time, FP is unlikely to be a better alternative to CT for reducing HR. Of all the HR data recorded, the mean HR shift was highest in the second half for FP (Table 5).

The evidence could have been affected by higher user immersion and engagement for FP. This could have led to participants exerting themselves more, thus, leading to higher HR readings. Other factors could be inconsistencies between the features of virtual avatars

and reality, a lack of control over movements or latency in responding. However, this condition is not visually invasive, with users mostly paying little attention to the avatar or member representation, suggesting that discomfort from the straps or the weight of the trackers could have had some impact.

The SSQ provided evidence that FP increased SS, and the objective data provided evidence that HR was increased for typical user behaviour. Overall, this suggests that FP is not an effective strategy for mitigating CS. Therefore, the hypothesis for this condition ($H_2$: Adding a first-person perspective with member representation to a VR experience mitigates CS) cannot be supported in the context of this experience, though it may offer benefits for other applications. However, for this, additional research is required.

5.2.4. Fans and Vibration

From applying Dunnett's test, no evidence of a significant difference between FV and CT was found for the total severity of SS. This was indicated by a *p*-value of $p = 0.7045$ (evidence against the rejection of the null hypothesis) and a confidence interval of the difference from CT between $-1.2244$ and $2.9908$ (including zero) (Table 3).

The effect of FV on HR appeared to be influenced by changes in all of the MVs. Compared to CT, FV had higher HR gradients that were positive for variations in distance travelled, speed and height above the ground with lower intercepts. It also had a lower HR gradient that was negative for variations in acceleration with a lower intercept and a higher HR gradient that was positive for variations in deceleration with the same intercept. This suggests that HR begins much lower than CT for distance travelled, speed and height above the ground, but can become higher at large values. However, while acceleration starts lower and deceleration starts the same as CT, both decrease HR at higher absolute values. This had some similarities to FP and was subject to what users spent most of the time doing, but had flatter gradients for distance travelled and speed in addition to substantially lower intercepts for all MVs. Of all the HR data recorded, the mean HR shift was lowest in both halves for FV (Table 5).

This result could have been impacted by our implementation of FV. The fans may have been too weak or not reactive enough and users could have been wearing clothes that covered their arms and legs. It could also be the case that the vibration was mismatched with the virtual experience. Activating the vibration based on flying speed and collisions may not have been appropriate for reducing a visual-vestibular sensory conflict from acceleration. It might have been important that participants could relate the feedback to personal experiences in the real world.

The SSQ provided no evidence that FV affects the level of SS and the objective results provided evidence that HR is mostly decreased but can be influenced by high-value MVs. Overall, these results are mixed. However, while it is clear that this condition has some impact on HR, more evidence would be needed on its link to CS for this particular use case to doubt the validity of the SSQ. Therefore, a judgement on the hypothesis for this condition ($H_3$: Simulating vibration and airflow in a VR experience mitigates CS) cannot be made without further investigation.

*5.3. Experiment Limitations*

5.3.1. Systematic Errors

While the SSQ is commonly used in research for assessing CS for VR applications, according to Stanney et al. [51] there is a distinction between SS and CS. This may mean the SSQ is not entirely suitable. It also does not consider a user physically interacting with a travel interface. Another consideration is that the SSQ is recommended to be administered post-exposure [51]. However, when electing to run an experiment between subjects, it does not provide an even comparison due to differences between subjects. Something like the Delta-SSQ used by Jung et al. [21] could have been more appropriate, comparing before and after exposure for each participant.

A disadvantage of comparing MVs against HR is that an instantaneous HR response is assumed. It could take time for a stimulus to have an effect on the user and for the PPG sensor to register a change. The PPG sensor is also limited in its accuracy to the nearest integer and produces an HR reading with a frequency of 2 Hz. This could have led to missing small fluctuations and identifying false relationships between variables. It could also have been the case that the level of inducement was too low to affect some people.

5.3.2. Random Errors

The main source of random error came from the sample of participants that signed up for the study. In such a group, many uncontrolled variables could have influenced the outcome of the experiment and how much the sample represented the entire population. This could include navigation path in the VE, physical exertion, body mass/height, demographic, experience with VR, acuity of eyesight, balance, interest, prior food/liquid consumption, phobias and clothing worn. Additionally, not everyone has the same change in HR with respect to their baselines for different stimuli. The baselines may also not have been representative of their resting HRs. Participants could have been nervous or exerted themselves before the experiment. Additionally, uncontrolled variables that could have affected the participants include heat, humidity, time of the day and lag in performance.

The participant data collected in the survey indicated variations in age, gender and experience with VR across conditions, factors known to influence CS. Most notably, all FV participants were male and there was a wide range of ages across all positively-skewed conditions. This compromises the generalisability of the results. In part, this can be attributed to the random assignment to conditions and the low total number of participants recruited. In a much larger study, similar distributions across conditions might have been attained. However, this was not possible due to the COVID-19 environment and random balancing before each experiment might have been more appropriate.

Despite applying transformations to the data, there remained three data sets from nausea, oculomotor and disorientation component analyses that did not have significant evidence of normality. This meant that some confidence was lost in the insight into nausea and oculomotor components from the ANOVA and Dunnett's tests. However, there was significant evidence that all of the conditions for total severity were normal.

The significance tests were susceptible to statistical error. Based on the chosen significance level, there was a 10% probability that the null hypothesis was falsely rejected in a Type I error for each test. There was also a chance that a null hypothesis failed to be rejected when it should have been in a Type II error.

*5.4. Significance of Findings*

From this experiment, evidence was found to suggest GV and FP are unsuccessful strategies to reduce CS. This indicates that users are sensitive to visual movements and representations. However, FV is a promising strategy. While it did not convincingly indicate that it does reduce CS according to the SSQ, it did have a significant impact on the experience as indicated by HR.

While the results do not indicate that CS has been successfully mitigated, the strategies still contribute to the techniques tried that dynamically manipulate FOV, include first-perspectives with member representation and have multi-sensory elements. These can still help to inform what is or is not included in developer guidelines and provide a foundation for further research.

**6. Conclusions**

This article presents an investigation of strategies that can be used to mitigate CS in VR using a novel travel interface. Three hypotheses for strategies to mitigate CS were devised. These were incorporated into an experiment with three manipulation conditions and a control condition. This involved designing a baseline experience, with a modified stool and a control system to navigate through a VE populated with models by flying. Each

Condition was designed and developed to be applied on top of this. During the experiment, physiological, flight data and camera footage were recorded for analysis. Post-exposure, a survey was also filled out by participants.

From conducting the experiment, the results suggest that GV and FP made SS worse with the HR from FP in most cases higher than CT. However, FV had mixed results. The SSQ found that adding these cues had no effect, yet the HR data was significantly lower for variations in MVs than in the other conditions. Ultimately, a larger, more focused investigation with a better balancing of demographics and further critique of the validity of the measures of CS would be required for more certainty in these results. However, despite some challenges with controlling the travel interface, users found the experience enjoyable, immersive and entertaining.

In future research, it would be interesting to see the effect of improving the responsiveness, comfort and stability of the travel interface based on user feedback. A more focused experiment could also be designed with a higher level of CS inducement where those who have extreme susceptibility to CS are excluded beforehand. This could increase the number of people who experience any symptoms and could remove outliers from the results.

More investigation could be done on FV to determine what causes the changes in HR, whether they are related to CS, and confirm if the SSQ result was correct. This could involve trying other physiological measures such as GSR, EEG or reaction time. Additionally, surveys for usability, presence, engagement and embodiment could help to understand the results.

**Author Contributions:** Conceptualization, D.P.; methodology, D.P.; investigation, D.P.; data curation, D.P.; supervision, R.W.L. and S.L.; writing—original draft preparation, D.P.; writing—review and editing, D.P., R.W.L. and S.L. All authors have read and agreed to the published version of the manuscript.

**Funding:** This research received no external funding.

**Institutional Review Board Statement:** The study was approved by the Human Ethics Committee of the University of Canterbury (Ref: HEC2021/69/LR, Granted: 13 September 2021).

**Informed Consent Statement:** Informed consent was obtained from all subjects involved in the study. Written informed consent has also been obtained from the subjects to publish this paper.

**Data Availability Statement:** The datasets generated and analysed during the current study are not publicly available due to the ethics application stating that the data will be destroyed five years after completion of the research.

**Conflicts of Interest:** The authors declare no conflict of interest.

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
