# Peer review of "Identifying Strategies to Mitigate Cybersickness in Virtual Reality Induced by Flying with an Interactive Travel Interface"

_mti, doi:10.3390/mti7050047_

Round 1
Reviewer 1 Report
The paper provides valuable insights into the factors that contribute to cybersickness in virtual reality (VR) environments and proposes potential strategies to mitigate its effects. The paper is well-structured, and the authors have provided a clear and concise overview of the experimental design, methodology, results, and conclusions.
One of the strengths of this paper is the use of physiological data to complement self-reported symptoms. This approach provides a more objective and accurate measure of the impact of different VR conditions on participants. The study's findings on the negative impact of GV and FP on cybersickness are significant and offer insights into the design of future VR experiences. Additionally, the paper's conclusion that FV may have some positive effect on the overall experience is noteworthy and suggests that further investigation is needed to explore its potential as a strategy for mitigating cybersickness.
The authors' attention to detail in describing the participants' experience and comments adds depth to the paper and makes it more relatable to readers. The study's sample size and randomized allocation of participants across conditions add to the paper's credibility, and the use of the Simulator Sickness Questionnaire as a reliable tool for measuring cybersickness further strengthens its validity.
The authors' findings and recommendations will be valuable to designers and developers of VR applications who seek to enhance user experience and mitigate the negative effects of cybersickness.
Author Response
Thank you for your feedback.
Reviewer 2 Report
1. A brief summary: The paper presents research on mitigation of cybersickness occurring while being in virtual reality. The paper identifies new cybersickness mitigation strategies.
2. General concept comments:
a) Weaknesses of the paper: the research trial (37 people) could be larger
b) Strengths of the paper: discussing new topic; topic of interaction of technology and people’s well-being – important topic in practice; finding solution for a real problem of cybersickness; well-structured paper; developed methodology
c) Hypotheses / goals / research gap: well-defined goals and hypotheses (H1, H2, H3)
d) Methodology: well-described methodology, research design (in details) and the whole experiment
e) Literature: proper selection, 51 references
3. Specific comments:
The article presents very interesting and original research. The whole experiment is well designed and well organized. The topic is new and up to date. The material and methods were prepared by the authors themselves. The experiment contributes to broadening new knowledge about the virtual reality environment and mitigation of cybersickness. The entire paper is done in a logical and structured way.
I recommend the publication of the article.
Author Response
Thank you for your feedback.
Reviewer 3 Report
The aim of this study was to design and test three Cybersickness mitigation strategies, including gaze tracking vignette, first-person perspective, and fans and vibration. The questionnaire data suggested that GV and FP induced a significant severe CS, whereas FV did not any changes in CS. However, the physiological results suggested that FV conditions showed a lower heart rate that may imply some effect on the experience.
I can see the cybersickness theories and measurement methods were reviewed in detail in section 2. I am not sure this level of detail is required for your paper. I think the focus of this study is your three mitigation strategies, which start from section 3. Worth to consider only keeping the important information related to your experiment design.
During the recruitment of subjects, is there a collection of their past history or experience with motion sickness or use of VR? As there is literature suggesting their past experience and expectation will play an important role in your results. Worth to add a discussion on this point.
Is there training needed for your subjects to understand how to use the kits? Is there a further analysis of the characteristics of subjects between the four groups. as we know there are gender and age effects on CS. Are you able to find a gender effect or a correlation between CS and age?
Interesting results on the CS and mitigation methods, I have noticed the large variation from your total severity scores (Figure 17), worth also looking at the effect size when there is a significant difference identified.
For the physiological measures, heart rate has been monitored, worth to reporting its baseline and then the shifting. What is the variation? Literature has suggested that heart rate is also related to difficulty in performing certain tasks. Is there a trend with time? E.g. the heart rate increase at the beginning and then decrease when subjects got used to the task or exposure.
You have observed an after-effect of exposure. E.g. some subjects showed an unsteady state after the end of exposure, any data has been collected on this. if not, worth to address this in your discussion. You have subjects’ verbally reported symptoms, may be worth adding what questions have you used to collect this feedback, e.g. symptom checklist or other methods.
Minor point:
Page 2, paragraph 2, an abbreviation of FOV, first use without explanation.
Worth to considering show the physiological data in a figure, so the trend could be easily compared.
Author Response
Point 1:
- Details not relevant to the mitigation strategies have been removed from S2. These were:
- Independent Visual Background (S2.3.1)
- Subjective Measures (S2.4.1; 168-181)
- Level of Control and Prediction Cues (S2.2.1; 83-91)
Point 2:
- There was no collection of their past history or experience during recruitment, but data was collected after each had done the experiment in the form of the survey, verbal remarks and observations by the experimenter.
- In recruitment those who knew that they had these issues and followed the screening advice would not have done this study (added to S3.8).
- Citations on building up a tolerance/adapting with experience to cybersickness added (S5.2).
Point 3:
- Instruction and “training”/familiarisation period described in S3.9.
- A table of correlations (Table 4) has been added for Gender vs SSQ total (point-biserial), Age vs SSQ total and VR Experience vs SSQ total (S4.2.4).
- A brief description of how these were computed (S3.6.5) and the significance (S5.2) added.
Point 4:
- Effect size (Cohen’s f) added (S4.2.3).
- How this was computed (S3.6.5) and its interpretation added (S5.2).
Point 5:
- A table has been added (Table 5) summarising (min/max/mean/sd) heart rate values at different stages of the experiment for the baseline, the first half (shift from baseline), and the second half (shift from baseline) split into conditions (S4.3).
- Details on this added in S3.6.5 and discussed in S5.2.
- Mention of mental/physical workload and heart rate/cybersickness added (S5.2).
Point 6:
- Additional information on what questions were in the survey contains added to S3.9.
- What participants were asked after the experiment has been added (S3.9).
- Brief mention of post-exposure state and lasting symptoms added (S4.4).
Point 7:
- First use abbreviation error amended (S1).
Point 8:
- Four figures were added (Figures 18-21) showing the physiological data (instead of a table of equations) with trend lines for comparison (S4.3).
Round 2
Reviewer 3 Report
All comments have been addressed in the revised manuscript